# Unsupervised Learning of Lagrangian Dynamics from Images for Prediction and Control

**Yaofeng Desmond Zhong**
Princeton University
y.zhong@princeton.edu

**Naomi Ehrich Leonard**
Princeton University
naomi@princeton.edu

## Abstract

Recent approaches for modelling dynamics of physical systems with neural networks enforce Lagrangian or Hamiltonian structure to improve prediction and generalization. However, when coordinates are embedded in high-dimensional data such as images, these approaches either lose interpretability or can only be applied to one particular example. We introduce a new unsupervised neural network model that learns Lagrangian dynamics from images, with interpretability that benefits prediction and control. The model infers Lagrangian dynamics on generalized coordinates that are simultaneously learned with a coordinate-aware variational autoencoder (VAE). The VAE is designed to account for the geometry of physical systems composed of multiple rigid bodies in the plane. By inferring interpretable Lagrangian dynamics, the model learns physical system properties, such as kinetic and potential energy, which enables long-term prediction of dynamics in the image space and synthesis of energy-based controllers.

## 1 Introduction

Humans can learn to predict the trajectories of mechanical systems, e.g., a basketball or a drone, from high-dimensional visual input, and learn to control the system, e.g., catch a ball or maneuver a drone, after a small number of interactions with those systems. We hypothesize that humans use domain-specific knowledge, e.g., physics laws, to achieve efficient learning. Motivated by this hypothesis, in this work, we propose incorporation of physics priors to learn and control dynamics from image data, aiming to gain interpretability and data efficiency. Specifically, we incorporate Lagrangian dynamics as the physic prior, which enables us to represent a broad class of physical systems. Recently, an increasing number of works [1, 2, 3, 4, 5] have incorporated Lagrangian/Hamiltonian dynamics into learning dynamical systems from coordinate data, to improve prediction and generalization. These approaches, however, require coordinate data, which are not always available in real-world applications. Hamiltonian Neural Network (HNN) [3] provides a single experiment with image observations, which requires a modification in the model. This modification is hard to generalize to systems with multiple rigid bodies. Hamiltonian Generative Network (HGN) [6] learns Hamiltonian dynamics from image sequences. However, the dimension of latent generalized coordinates is $4 \times 4 \times 16 = 256$, making interpretation difficult. Moreover, both HNN and HGN focus on prediction and have no design of control. Another class of approaches learn physical models from images, by either learning the map from images to coordinates with supervision on coordinate data [7] or learning the coordinates in an unsupervised way but only with translational coordinates [8, 9]. The unsupervised learning of rotational coordinates such as angles are under-explored in the literature.

In this work, we propose an unsupervised neural network model that learns coordinates and Lagrangian dynamics on those coordinates from images of physical systems in motion in the plane. The latent dynamical model enforces Lagrangian dynamics, which benefits long term prediction of the system. As Lagrangian dynamics commonly involve rotational coordinates to describe the changing

configurations of objects in the system, we propose a coordinate-aware variational autoencoder (VAE) that can infer interpretable rotational and translational coordinates from images without supervision. The interpretable coordinates together with the interpretable Lagrangian dynamics pave the way for introducing energy-based controllers of the learned dynamics.

## 1.1 Related work

**Lagrangian/Hamiltonian prior in learning dynamics**    To improve prediction and generalization of physical system modelling, a class of approaches has incorporated the physics prior of Hamiltonian or Lagrangian dynamics into deep learning. Deep Lagrangian Network [1] and Lagrangian Neural Network [2] learn Lagrangian dynamics from position, velocity and acceleration data. Hamiltonian Neural Networks [3] learn Hamiltonian dynamics from position, velocity and acceleration data. By leveraging ODE integrators, Hamiltonian Graph Networks [10] and Symplectic ODE-Net [4] learn Hamiltonian dynamics from only position and velocity data. All of these works (except one particular experiment in HNN [3]) require direct observation of low dimensional position and velocity data. Hamiltonian Generative Network [6] learns Hamiltonian dynamics from images.

**Unsupervised learning of dynamics**    We assume we are given no coordinate data and aim to learn coordinates and dynamics in an unsupervised way. With little position and velocity data, Belbute-Peres et al. [11] learn underlying dynamics. However, the authors observe that their model fails to learn meaningful dynamics when there is no supervision on position and velocity data at all. Without supervision, Watter et al. [12] and Levine et al. [13] learn locally linear dynamics and Jaques et al. [14] learns unknown parameters in latent dynamics with a given form. Kossen et al. [15] extracts position and velocity of each object from videos and learns the underlying dynamics. Watters et al. [16] adopts an object-oriented design to gain data efficiency and robustness. Battaglia et al. [17], Sanchez-Gonzalez et al. [18] and Watters et al. [7] learn dynamics with supervision by taking into account the prior of objects and their relations. These object-oriented designs focus little on rotational coordinates. Variational Integrator Network [19] considers rotational coordinates but cannot handle systems with multiple rotational coordinates.

**Neural visual control**    Besides learning dynamics for prediction, we would like to learn how control input influences the dynamics and to design control laws based on the learned model. This goal is relevant to neural motion planning and model-based reinforcement learning from images. PlaNet [20] learns latent dynamical models from images and designs control input by fast planning in the latent space. Kalman VAE [21] can potentially learn locally linear dynamics and control from images, although no control result has been shown. Dreamer [22] is a scalable reinforcement learning agent which learns from images using a world model. Ebert et al. [23] propose a self-supervised model-based method for robotic control. We leave the comparison of our energy-based control methods and these model-based control methods in the literature to future work.

## 1.2 Contribution

The main contribution of this work is two-fold. First, we introduce an unsupervised learning framework to learn Lagrangian dynamics from image data for prediction. The Lagrangian prior conserves energy with no control applied; this helps learn more accurate dynamics as compared to a MLP dynamical model. Moreover, the coordinate-aware VAE in the proposed learning framework infers interpretable latent rigid body coordinates in the sense that a coordinate encoded from an image of a system in a position with high potential energy has a high learned potential energy, and vice versa. This interpretability enables us to design energy-based controllers to control physical systems to target positions. We implement this work with PyTorch [24] and refactor our code into PyTorch Lightning format [25], which makes our code easy to read and our results easy to reproduce. The code for all experiments is available at `https://github.com/DesmondZhong/Lagrangian_caVAE`.

## 2 Preliminary concepts

### 2.1 Lagrangian dynamics

Lagrangian dynamics are a reformulation of Newton's second law of motion. The configuration of a system in motion at time $t$ is described by generalized coordinates $\mathbf{q}(t) = (q_1(t), q_2(t), ..., q_m(t))$,

where $m$ is the number of degrees of freedom (DOF) of the system. For planar rigid body systems with $n$ rigid bodies and $k$ holonomic constraints, the DOF is $m = 3n - k$. From D'Alembert's principle, the equations of motion of the system, also known as the Euler-Lagrange equation, are

$$\frac{\mathrm{d}}{\mathrm{d}t}\left(\frac{\partial L}{\partial \dot{\mathbf{q}}}\right) - \frac{\partial L}{\partial \mathbf{q}} = \mathbf{Q}^{nc}, \tag{1}$$

where the scalar function $L(\mathbf{q}, \dot{\mathbf{q}})$ is the Lagrangian, $\dot{\mathbf{q}} = \mathrm{d}\mathbf{q}/\mathrm{d}t$, and $\mathbf{Q}^{nc}$ is a vector of non-conservative generalized forces. The Lagrangian $L(\mathbf{q}, \dot{\mathbf{q}})$ is the difference between kinetic energy $T(\mathbf{q}, \dot{\mathbf{q}})$ and potential energy $V(\mathbf{q})$. For rigid body systems, the Lagrangian is

$$L(\mathbf{q}, \dot{\mathbf{q}}) = T(\mathbf{q}, \dot{\mathbf{q}}) - V(\mathbf{q}) = \frac{1}{2}\dot{\mathbf{q}}^T \mathbf{M}(\mathbf{q})\dot{\mathbf{q}} - V(\mathbf{q}), \tag{2}$$

where $\mathbf{M}(\mathbf{q})$ is the mass matrix. In this work, we assume that the control inputs are the only non-conservative generalized forces, i.e., $\mathbf{Q}^{nc} = \mathbf{g}(\mathbf{q})\mathbf{u}$, where $\mathbf{g}(\mathbf{q})$ is the input matrix and $\mathbf{u}$ is a vector of control inputs such as forces or torques. Substituting $\mathbf{Q}^{nc} = \mathbf{g}(\mathbf{q})\mathbf{u}$ and $L(\mathbf{q}, \dot{\mathbf{q}})$ from (2) into (1), we get the equations of motion in the form of $m$ second-order ordinary differential equations (ODE):

$$\ddot{\mathbf{q}} = \mathbf{M}^{-1}(\mathbf{q})\left(-\frac{1}{2}\frac{\mathrm{d}\mathbf{M}(\mathbf{q})}{\mathrm{d}t}\dot{\mathbf{q}} - \frac{\mathrm{d}V(\mathbf{q})}{\mathrm{d}\mathbf{q}} + \mathbf{g}(\mathbf{q})\mathbf{u}\right). \tag{3}$$

## 2.2 Control via energy shaping

Our goal is to control the system to a reference configuration $\mathbf{q}^\star$, inferred from a goal image $\mathbf{x}^\star$, based on the learned dynamics. As we are essentially learning the kinetic and potential energy associated with the system, we can leverage the learned energy for control by *energy shaping* [26, 27].

If $\mathrm{rank}(\mathbf{g}(\mathbf{q})) = m$, we have control over every DOF and the system is fully actuated. For such systems, control to the reference configuration $\mathbf{q}^\star$ can be achieved with the control law $\mathbf{u}(\mathbf{q}, \dot{\mathbf{q}}) = \boldsymbol{\beta}(\mathbf{q}) + \mathbf{v}(\dot{\mathbf{q}})$, where $\boldsymbol{\beta}(\mathbf{q})$ is the *potential energy shaping* and $\mathbf{v}(\dot{\mathbf{q}})$ is the *damping injection*. The goal of potential energy shaping is to let the system behave as if it is governed by a desired Lagrangian $L_d$ with no non-conservative generalized forces.

$$\frac{\mathrm{d}}{\mathrm{d}t}\left(\frac{\partial L}{\partial \dot{\mathbf{q}}}\right) - \frac{\partial L}{\partial \mathbf{q}} = \mathbf{g}(\mathbf{q})\boldsymbol{\beta}(\mathbf{q}) \quad \Longleftrightarrow \quad \frac{\mathrm{d}}{\mathrm{d}t}\left(\frac{\partial L_d}{\partial \dot{\mathbf{q}}}\right) - \frac{\partial L_d}{\partial \mathbf{q}} = 0, \tag{4}$$

where the desired Lagrangian has desired potential energy $V_d(\mathbf{q})$:

$$L_d(\mathbf{q}, \dot{\mathbf{q}}) = T(\mathbf{q}, \dot{\mathbf{q}}) - V_d(\mathbf{q}) = \frac{1}{2}\dot{\mathbf{q}}^T \mathbf{M}(\mathbf{q})\dot{\mathbf{q}} - V_d(\mathbf{q}). \tag{5}$$

The difference between $L_d$ and $L$ is the difference between $V$ and $V_d$, which explains the name potential energy shaping: $\boldsymbol{\beta}(\mathbf{q})$ shapes the potential energy $V$ of the original system into a desired potential energy $V_d$. The potential energy $V_d$ is designed to have a global minimum at $\mathbf{q}^\star$. By the equivalence (4), we get

$$\boldsymbol{\beta}(\mathbf{q}) = \mathbf{g}^T(\mathbf{g}\mathbf{g}^T)^{-1}\left(\frac{\partial V}{\partial \mathbf{q}} - \frac{\partial V_d}{\partial \mathbf{q}}\right). \tag{6}$$

With only potential energy shaping, the system dynamics will oscillate around $\mathbf{q}^\star$.[1] The purpose of damping injection $\mathbf{v}(\dot{\mathbf{q}})$ is to impose convergence, exponentially in time, to $\mathbf{q}^\star$. The damping injection has the form

$$\mathbf{v}(\dot{\mathbf{q}}) = -\mathbf{g}^T(\mathbf{g}\mathbf{g}^T)^{-1}(\mathbf{K}_d\dot{\mathbf{q}}). \tag{7}$$

For underactuated systems, however, this controller design is not valid since $\mathbf{g}\mathbf{g}^T$ will not be invertible. In general, we also need kinetic energy shaping [27] to achieve a control goal.

**Remark** The design parameters here are $V_d$ and $\mathbf{K}_d$. A quadratic desired potential energy

$$V_d(\mathbf{q}) = \frac{1}{2}(\mathbf{q} - \mathbf{q}^\star)^T \mathbf{K}_p(\mathbf{q} - \mathbf{q}^\star), \tag{8}$$

results in a controller design

$$\mathbf{u}(\mathbf{q}, \dot{\mathbf{q}}) = \mathbf{g}^T(\mathbf{g}\mathbf{g}^T)^{-1}\left(\frac{\partial V}{\partial \mathbf{q}} - \mathbf{K}_p(\mathbf{q} - \mathbf{q}^\star) - \mathbf{K}_d\dot{\mathbf{q}}\right). \tag{9}$$

This can be interpreted as a proportional-derivative (PD) controller with energy compensation.

## 2.3 Training Neural ODE with constant control

The Lagrangian dynamics can be formulated as a set of first-order ODE

$$\dot{\mathbf{s}} = \mathbf{f}(\mathbf{s}, \mathbf{u}), \tag{10}$$

where $\mathbf{s}$ is a state vector and unknown vector field $\mathbf{f}$, which is a vector-valued function, can be parameterized with a neural network $\mathbf{f}_\psi$. We leverage Neural ODE, proposed by Chen et al. [28], to learn the function $\mathbf{f}$ that explains the trajectory data of $\mathbf{s}$. The idea is to predict future states from an initial state by integrating the ODE with an ODE solver. As all the operations in the ODE solver are differentiable, $\mathbf{f}_\psi$ can be updated by back-propagating through the ODE solver and approximating the true $\mathbf{f}$. However, Neural ODE cannot be applied to (10) directly since the input dimension and the output dimension of $\mathbf{f}$ are not the same. Zhong et al. [4] showed that if the control remains constant for each trajectory in the training data, Neural ODE can be applied to the following augmented ODE:

$$\begin{pmatrix} \dot{\mathbf{s}} \\ \dot{\mathbf{u}} \end{pmatrix} = \begin{pmatrix} \mathbf{f}_\psi(\mathbf{s}, \mathbf{u}) \\ \mathbf{0} \end{pmatrix} = \tilde{\mathbf{f}}_\psi(\mathbf{s}, \mathbf{u}). \tag{11}$$

With a learned $\mathbf{f}_\psi$, we can apply a controller design $\mathbf{u} = \mathbf{u}(\mathbf{s})$ that is not constant, e.g., an energy-based controller, by integrating the ODE $\dot{\mathbf{s}} = \mathbf{f}(\mathbf{s}, \mathbf{u}(\mathbf{s}))$.

# 3 Model architecture

Let $\mathbf{X} = ((\mathbf{x}^0, \mathbf{u}^c), (\mathbf{x}^1, \mathbf{u}^c)), ..., (\mathbf{x}^{T_{\text{pred}}}, \mathbf{u}^c))$ be a given sequence of image and control pairs, where $\mathbf{x}^\tau, \tau = 0, 1, \ldots, T_{\text{pred}}$, is the image of the trajectory of a rigid-body system under constant control $\mathbf{u}^c$ at time $t = \tau \Delta t$. From $\mathbf{X}$ we want to learn a state-space model (10) that governs the time evolution of the rigid-body system dynamics. We assume the number of rigid bodies $n$ is known and the segmentation of each object in the image is given. Each image can be written as $\mathbf{x}^\tau = (\mathbf{x}_1^\tau, ..., \mathbf{x}_n^\tau)$, where $\mathbf{x}_i^\tau \in \mathbb{R}^{n_x}$ contains visual information about the $i$th rigid body at $t = \tau \Delta t$ and $n_x$ is the dimension of the image space.

In Section 3.1, we parameterize $\mathbf{f}(\mathbf{s}, \mathbf{u})$ with a neural network and design the architecture of the neural network such that (10) is constrained to follow Lagrangian dynamics, where the physical properties such as mass and potential energy are learned from data. Since we have no access to state data, we need to infer states $\mathbf{s}$, i.e., generalized coordinates and velocities from image data. Sections 3.2 and 3.4 introduce an inference model (encoder) and a generative model (decoder) pair. Together they make up a variational autoencoder (VAE) [29] to infer the generalized coordinates in an unsupervised way. Section 3.3 introduces a simple estimator of velocity from learned generalized coordinates. The VAE and the state-space model are trained together, as described in Section 3.5. The model architecture is shown in Figure 1.

## 3.1 Latent Lagrangian dynamics

The Lagrangian dynamics (3) yield a second-order ODE. From a model-based perspective, they can be re-written as a first-order ODE (10) by choosing the state as $\mathbf{s} = (\mathbf{q}, \dot{\mathbf{q}})$. However, from a data-driven perspective, this choice of state is problematic when the generalized coordinates involve angles. Consider the pendulum task in Figure 2 as an example where we want to infer the generalized coordinate, i.e., the angle of the pendulum $\phi$, from an image of the pendulum. The map from the image to the angle $\phi$ should be bijective. However, if we choose the state as $\mathbf{s} = (\phi, \dot{\phi})$, the map is not bijective, since $\phi$ and $\phi + 2\pi$ map to the same image. If we restrict $\phi \in [-\pi, \pi)$, then the dynamics are not continuous when the pendulum moves around the inverted position. Inspired by Zhong et al. [4], we solve this issue by proposing the state as $\mathbf{s} = (\cos\phi, \sin\phi, \dot{\phi})$, such that the mapping from the pendulum image to $(\cos\phi, \sin\phi)$ is bijective.

In general, for a planar rigid-body system with $\mathbf{q} = (\mathbf{r}, \boldsymbol{\phi})$, where $\mathbf{r} \in \mathbb{R}^{m_R}$ are translational generalized coordinates and $\boldsymbol{\phi} \in \mathbb{T}^{m_T}$ are rotational generalized coordinates , the proposed state is $\mathbf{s} = (\mathbf{s}_1, \mathbf{s}_2, \mathbf{s}_3, \mathbf{s}_4, \mathbf{s}_5) = (\mathbf{r}, \cos\boldsymbol{\phi}, \sin\boldsymbol{\phi}, \dot{\mathbf{r}}, \dot{\boldsymbol{\phi}})$, where $\cos$ and $\sin$ are applied element-wise to $\boldsymbol{\phi}$. To enforce Lagrangian dynamics in the state-space model, we take the derivative of $\mathbf{s}$ with respect to $t$

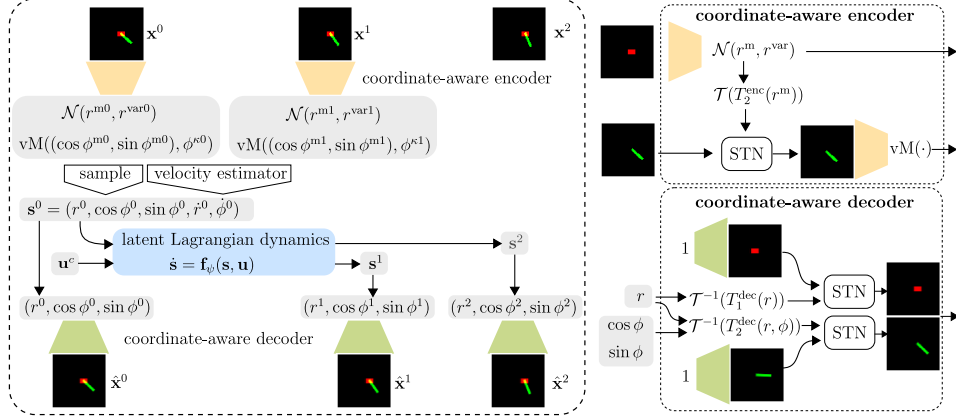

Figure 1: **Left**: Model architecture. (Using CartPole as an illustrative example.) The initial state $\mathbf{s}^0$ is constructed by sampling the distribution and a velocity estimator. The latent Lagrangian dynamics take $\mathbf{s}^0$ and the constant control $\mathbf{u}^c$ for that trajectory and predict future states up to $T_{\text{pred}}$. The diagram shows the $T_{\text{pred}} = 2$ case. **Top-right**: The coordinate-aware encoder estimates the distribution of generalized coordinates. **Bottom-right**: The initial and predicted generalized coordinates are decoded to the reconstruction images with the coordinate-aware decoder.

and substitute in (3) to get

$$
\dot{\mathbf{s}} = \begin{pmatrix} \mathbf{s}_4 \\ -\mathbf{s}_3 \circ \mathbf{s}_5 \\ \mathbf{s}_2 \circ \mathbf{s}_5 \\ \mathbf{M}^{-1}(\mathbf{s}_1,\mathbf{s}_2,\mathbf{s}_3)\left(-\frac{1}{2}\frac{\mathrm{d}\mathbf{M}(\mathbf{s}_1\mathbf{s}_2,\mathbf{s}_3)}{\mathrm{d}t}\begin{pmatrix}\mathbf{s}_4\\\mathbf{s}_5\end{pmatrix} + \begin{pmatrix}-\frac{\partial V(\mathbf{s}_1\mathbf{s}_2\mathbf{s}_3)}{\partial \mathbf{s}_1}\\ \frac{\partial V(\mathbf{s}_1\mathbf{s}_2\mathbf{s}_3)}{\partial \mathbf{s}_2}\mathbf{s}_3 - \frac{\partial V(\mathbf{s}_1\mathbf{s}_2\mathbf{s}_3)}{\partial \mathbf{s}_3}\mathbf{s}_2\end{pmatrix} + \mathbf{g}(\mathbf{s}_1,\mathbf{s}_2,\mathbf{s}_3)\mathbf{u}\right) \end{pmatrix}
\tag{12}
$$

where $\circ$ is the element-wise product. We use three neural networks, $\mathbf{M}_{\psi_1}(\mathbf{s}_1, \mathbf{s}_2, \mathbf{s}_3)$, $V_{\psi_2}(\mathbf{s}_1, \mathbf{s}_2, \mathbf{s}_3)$, and $\mathbf{g}_{\psi_3}(\mathbf{s}_1, \mathbf{s}_2, \mathbf{s}_3)$, to approximate the mass matrix, the potential energy and the input matrix, respectively. Equation (12) is then a state-space model parameterized by a neural network $\dot{\mathbf{s}} = \mathbf{f}_\psi(\mathbf{s}, \mathbf{u})$. It can be trained as stated in Section 2.3 given the initial condition $\mathbf{s}^0 = (\mathbf{r}^0, \cos\boldsymbol{\phi}^0, \sin\boldsymbol{\phi}^0, \dot{\mathbf{r}}^0, \dot{\boldsymbol{\phi}}^0)$ and $\mathbf{u}^c$. Next, we present the means to infer $\mathbf{s}^0$ from the given images.

### 3.2 Coordinate-aware encoder

From a latent variable modelling perspective, an image $\mathbf{x}$ of a rigid-body system can be generated by first specifying the values of the generalized coordinates and then assigning values to pixels based on the generalized coordinates with a generative model - the decoder. In order to infer those generalized coordinates from images, we need an inference model - the encoder. We perform variational inference with a coordinate-aware VAE.

The coordinate-aware encoder infers a distribution on the generalized coordinates. The Gaussian distribution is the default for modelling latent variables in VAE. This is appropriate for modelling a translational generalized coordinate $r$ since $r$ resides in $\mathbb{R}^1$. However, this is not appropriate for modelling a rotational generalized coordinate $\phi$ since a Gaussian distribution is not a distribution on $\mathbb{S}^1$. If we use a Gaussian distribution to model hyperspherical latent variables, the VAE performs worse than a traditional autoencoder [30]. Thus, to model $\phi$, we use the von Mises (vM) distribution, a family of distributions on $\mathbb{S}^1$. Analogous to a Gaussian distribution, a von Mises distribution is characterized by two parameters: $\mu \in \mathbb{R}^2$, $||\mu||^2 = 1$ is the mean, and $\kappa \in \mathbb{R}_{\geq 0}$ is the concentration around $\mu$. The von Mises distribution reduces to a uniform distribution when $\kappa = 0$.

In our model, for a rotational generalized coordinate $\phi$, we assume a posterior distribution $Q(\phi|\mathbf{x}) = \text{vM}((\cos\phi^{\text{m}}, \sin\phi^{\text{m}}), \phi^\kappa)$ with prior $P(\phi) = \text{vM}(\cdot, 0) = U(\mathbb{S}^1)$. For a translational generalized coordinate $r$, we assume a posterior distribution $Q(r|\mathbf{x}) = \mathcal{N}(r^{\text{m}}, r^{\text{var}})$ with prior $\mathcal{N}(0, 1)$. We denote the joint posterior distribution as $Q(\mathbf{q}|\mathbf{x})$ and joint prior distribution as $P(\mathbf{q})$. The encoder is a neural network that takes an image as input and provides the parameters of the distributions as output.

A black-box neural network encoder would not be able to learn interpretable generalized coordinates for a system in motion described by Lagrangian dynamics. Instead, we propose a coordinate-aware encoder by designing the architecture of the neural network to account for the geometry of the system. This is the key to interpretable encoding of generalized coordinates.

Recall that each generalized coordinate $q_j$ specifies the position/rotation of a rigid body $i_j$ in the system. In principle, the coordinate can be learned from the image segmentation of $i_j$. However, the reference frame of a generalized coordinate might depend on other generalized coordinates and change across images. Take the CartPole example in Figure 2 as motivation. The system has two DOF and

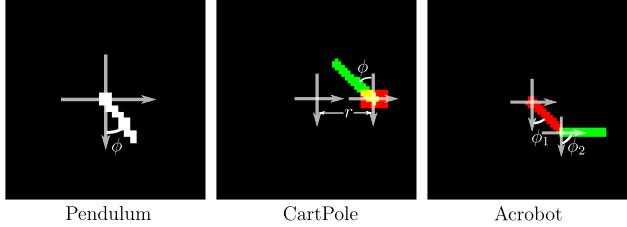

Figure 2: One choice of generalized coordinates and their corresponding reference frames in three example systems

natural choices of generalized coordinates are the horizontal position of the cart $q_1 = r$ and the angle of the pole $q_2 = \phi$. The origin of the reference frame of $r$ is the center of the image, which is the same across all images. The origin of the reference frame of $\phi$, however, is the center of the cart, which is not the same across all the images since the cart can move. In order to learn the angle of the pole, we can either use a translation invariant architecture such as Convolution Neural Networks (CNN) or place the center of the encoding attention window of the pole segmentation image at the center of the cart. The former approach does not work well in extracting generalized coordinates.[2] Thus, we adopt the latter approach, where we shift our encoding attention window horizontally with direction and magnitude given by generalized coordinate $r$, before feeding it into a neural network to learn $\phi$. In this way we exploit the geometry of the system in the encoder.

The default attention window is the image grid and corresponds to the default reference frame, where the origin is at the center of the image with horizontal and vertical axes. The above encoding attention window mechanism for a general system can be formalized by considering the transformation from the default reference frame to the reference frame of each generalized coordinate. The transformation of a point $(x_d, y_d)$ in the default reference frame to a point $(x_t, y_t)$ in the target reference frame is captured by transformation $\mathcal{T}(x, y, \theta)$ corresponding to translation by $(x, y)$ and rotation by $\theta$ as follows:

$$\begin{pmatrix} x_t \\ y_t \\ 1 \end{pmatrix} = \mathcal{T}(x, y, \theta) \begin{pmatrix} x_d \\ y_d \\ 1 \end{pmatrix}, \quad \text{where} \quad \mathcal{T}(x, y, \theta) = \begin{pmatrix} \cos\theta & \sin\theta & x \\ -\sin\theta & \cos\theta & y \\ 0 & 0 & 1 \end{pmatrix}. \tag{13}$$

So let $\mathcal{T}((x, y, \theta)_j^{\text{enc}})$ be the transformation from default frame to reference frame of generalized coordinate $q_j$. This transformation might depend on constant parameters $\mathbf{c}$ associated with the shape and size of the rigid bodies and generalized coordinates $\mathbf{q}_{-j}$, which denotes the vector of generalized coordinates with $q_j$ removed. Let $(x, y, \theta)_j^{\text{enc}} = T_j^{\text{enc}}(\mathbf{q}_{-j}, \mathbf{c})$. Both $\mathbf{q}_{-j}$ and $\mathbf{c}$ are learned from images. However, the function $T_j^{\text{enc}}$ is specified by leveraging the geometry of the system. In the CartPole example, $(q_1, q_2) = (r, \phi)$, and $T_1^{\text{enc}} \equiv (0, 0, 0)$ and $T_2^{\text{enc}}(q_1) = (q_1, 0, 0)$. In the Acrobot example, $(q_1, q_2) = (\phi_1, \phi_2)$, and $T_1^{\text{enc}} \equiv (0, 0, 0)$ and $T_2^{\text{enc}}(q_1, l_1) = (l_1 \sin q_1, l_1 \cos q_1, 0)$.

The shift of attention window can be implemented with a spatial transformer network (STN) [32], which generates a transformed image $\tilde{\mathbf{x}}_{i_j}$ from $\mathbf{x}_{i_j}$, i.e., $\tilde{\mathbf{x}}_{i_j} = \text{STN}(\mathbf{x}_{i_j}, \mathcal{T}(T_j^{\text{enc}}(\mathbf{q}_{-j}, \mathbf{c})))$. In general, to encode $q_j$, we use a multilayer perceptron (MLP) that takes $\tilde{\mathbf{x}}_{i_j}$ as input and provides the parameters of the $q_j$ distribution as output. For a translational coordinate $q_j$, we have $(q_j^{\text{m}}, \log q_j^{\text{var}}) = \text{MLP}_j^{\text{enc}}(\tilde{\mathbf{x}}_{i_j})$. For a rotational coordinate $q_j$, we have $(\alpha_j, \beta_j, \log q_j^\kappa) = \text{MLP}_j^{\text{enc}}(\tilde{\mathbf{x}}_{i_j})$, where the mean of the von Mises distribution is computed as $(\cos q_j^m, \sin q_j^m) = (\alpha_j, \beta_j)/\sqrt{\alpha_j^2 + \beta_j^2}$. We then

take a sample from the $q_j$ distribution.[3] Doing this for every generalized coordinate $q_j$, we can get $(\mathbf{r}^\tau, \cos\boldsymbol{\phi}^\tau, \sin\boldsymbol{\phi}^\tau)$ from $\mathbf{x}^\tau$ for any $\tau$.[4] We will use $(\mathbf{r}^0, \cos\boldsymbol{\phi}^0, \sin\boldsymbol{\phi}^0)$ and $(\mathbf{r}^1, \cos\boldsymbol{\phi}^1, \sin\boldsymbol{\phi}^1)$.

### 3.3 Velocity estimator

To integrate Equation (12), we also need to infer $(\dot{\mathbf{r}}^0, \dot{\boldsymbol{\phi}}^0)$, the initial velocity. We can estimate the initial velocity from the encoded generalized coordinates by finite difference. We use the following simple first-order finite difference estimator:

$$\dot{\mathbf{r}}^0 = (\mathbf{r}^{\text{m1}} - \mathbf{r}^{\text{m0}})/\Delta t, \tag{14}$$

$$\dot{\boldsymbol{\phi}}^0 = \left((\sin\boldsymbol{\phi}^{\text{m1}} - \sin\boldsymbol{\phi}^{\text{m0}}) \circ \cos\boldsymbol{\phi}^{\text{m0}} - (\cos\boldsymbol{\phi}^{\text{m1}} - \cos\boldsymbol{\phi}^{\text{m0}}) \circ \sin\boldsymbol{\phi}^{\text{m0}}\right)/\Delta t, \tag{15}$$

where $(\mathbf{r}^{\text{m0}}, \cos\boldsymbol{\phi}^{\text{m0}}, \sin\boldsymbol{\phi}^{\text{m0}})$ and $(\mathbf{r}^{\text{m1}}, \cos\boldsymbol{\phi}^{\text{m1}}, \sin\boldsymbol{\phi}^{\text{m1}})$ are the means of the generalized coordinates encoded from the image at time $t = 0$ and $t = \Delta t$, respectively. Jaques et al. [14] proposed to use a neural network to estimate velocity. From our experiments, our simple estimator works better than a neural network estimator.

### 3.4 Coordinate-aware decoder

The decoder provides a distribution $P(\mathbf{x}|\mathbf{q}) = \mathcal{N}(\hat{\mathbf{x}}, \mathbf{I})$ as output, given a generalized coordinate $\mathbf{q}$ as input, where the mean $\hat{\mathbf{x}}$ is the reconstruction image of the image data $\mathbf{x}$. Instead of using a black box decoder, we propose a coordinate-aware decoder. The coordinate-aware decoder first generates a static image $\mathbf{x}_i^c$ of every rigid body $i$ in the system, at a default position and orientation, using a MLP with a constant input, i.e., $\mathbf{x}_i^c = \text{MLP}_i^{\text{dec}}(1)$. The coordinate-aware decoder then determines $\hat{\mathbf{x}}_i$, the image of rigid body $i$ positioned and oriented on the image plane according to the generalized coordinates. The proposed decoder is inspired by the coordinate-consistent decoder by Jaques et al. [14]. However, the decoder of [14] cannot handle a system of multiple rigid bodies with constraints such as the Acrobot and the CartPole, whereas our coordinate-aware decoder can.

As in Jaques et al. [14], to find $\hat{\mathbf{x}}_i$ we use the inverse transformation matrix $\mathcal{T}^{-1}((x, y, \theta)_i^{\text{dec}})$ where $\mathcal{T}$ is given by (13) and $(x, y, \theta)_i^{\text{dec}} = T_i^{\text{dec}}(\mathbf{q}, \mathbf{c})$. In the CartPole example, $(q_1, q_2) = (r, \phi)$, and $T_1^{\text{dec}}(r) = (r, 0, 0)$ and $T_2^{\text{dec}}(r, \phi) = (r, 0, \phi)$. In the Acrobot example, $(q_1, q_2) = (\phi_1, \phi_2)$, and $T_1^{\text{dec}}(\phi_1) = (0, 0, \phi_1)$ and $T_2^{\text{dec}}(\phi_1, \phi_2) = (l_1 \sin\phi_1, l_1 \cos\phi_1, \phi_2)$. The reconstruction image is then $\hat{\mathbf{x}} = (\hat{\mathbf{x}}_1, ..., \hat{\mathbf{x}}_n)$, where $\hat{\mathbf{x}}_i = \text{STN}(\mathbf{x}_i^c, \mathcal{T}^{-1}(T_i^{\text{dec}}(\mathbf{q}, \mathbf{c})))$.

### 3.5 Loss function

The loss $\mathcal{L}(\mathbf{X})$ consists of the sum of three terms:

$$\mathcal{L}(\mathbf{X}) = \underbrace{-\mathbb{E}_{\mathbf{q}^0 \sim Q}[\log P(\mathbf{x}^0|\mathbf{q}^0)] + \text{KL}(Q(\mathbf{q}^0|\mathbf{x}^0)||P(\mathbf{q}^0))}_{\text{VAE loss}} + \underbrace{\sum_{\tau=1}^{T_{\text{pred}}} ||\hat{\mathbf{x}}^\tau - \mathbf{x}^\tau||_2^2}_{\text{prediction loss}} + \underbrace{\lambda \sum_j \sqrt{\alpha_j^2 + \beta_j^2}}_{\text{vM regularization}}. \tag{16}$$

The VAE loss is a variational bound on the marginal log-likelihood of initial data $P(\mathbf{x}^0)$. The prediction loss captures inaccurate predictions of the latent Lagrangian dynamics. The vM regularization with weight $\lambda$ penalizes large norms of vectors $(\alpha_j, \beta_j)$, preventing them from blowing up.

## 4 Results

We train our model on three systems: the Pendulum, the fully-actuated CartPole and the fully-actuated Acrobot. The training images are generated by OpenAI Gym simulator [33]. The training setup is detailed in Supplementary Materials. As the mean square error in the image space is not a good metric

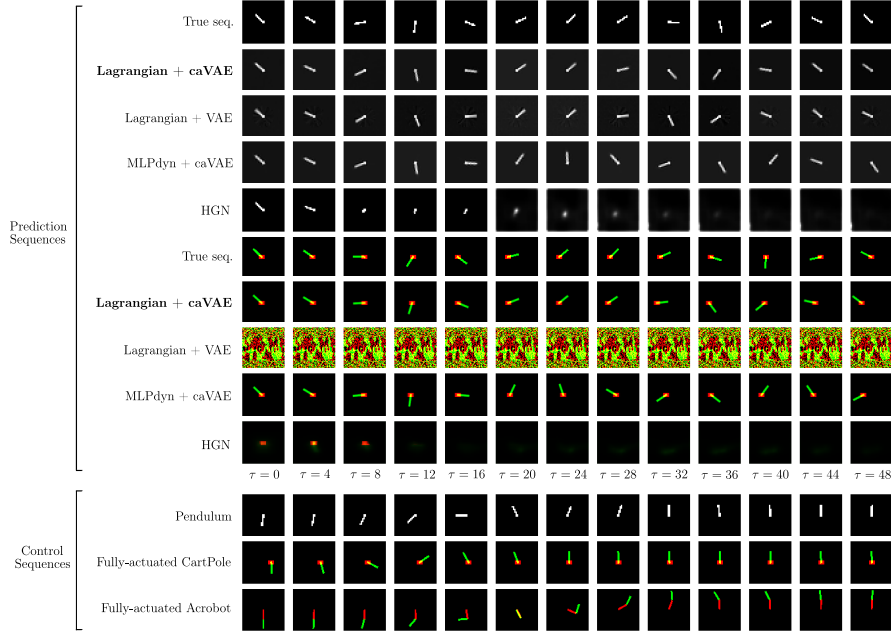

**Figure 3:** **Top**: Prediction sequences of Pendulum and CartPole with a previously unseen initial condition and zero control. Prediction results show both Lagrangian dynamics and coordinate-aware VAE are necessary to perform long term prediction. **Bottom**: Control sequences of three systems. Energy-based controllers are able to control the systems to the goal positions based on learned dynamics and encoding with Lagrangian+caVAE.

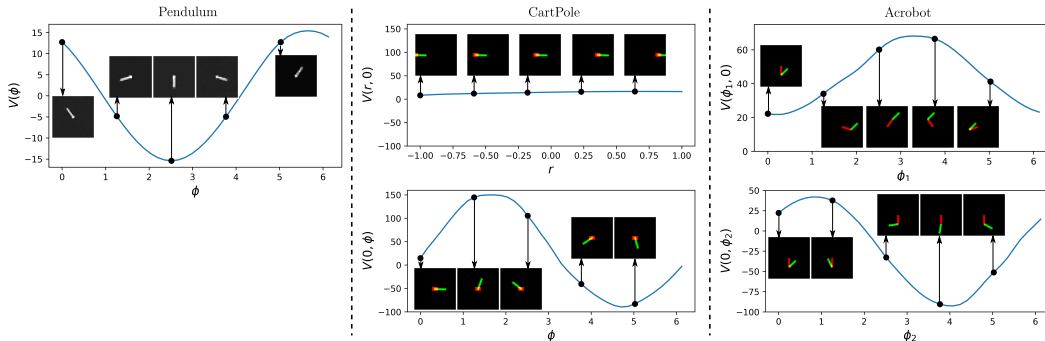

**Figure 4:** Learned potential energy with Lagrangian+caVAE of three systems and reconstruction images at selected coordinates. Both the learned coordinates and potential energy are interpretable.

of long term prediction accuracy [8], we report on the prediction image sequences of a previously unseen initial condition and highlight the interpretability of our model.

**Lagrangian dynamics and coordinate-aware VAE improve prediction.** As the Acrobot is a chaotic system, accurate long term prediction is impossible. Figure 3 shows the prediction sequences of images up to 48 time steps of the Pendulum and CartPole experiments with models trained with $T_{\text{pred}} = 4$. We compare the prediction results of our model (labelled as *Lagrangian+caVAE*) with two model variants: *MLPdyn+caVAE*, which replaces the Lagrangian latent dynamics with MLP latent dynamics, and *Lagrangian+VAE*, which replaces the coordinate-aware VAE with a traditional VAE. The traditional VAE fails to reconstruct meaningful images for CartPole, although it works well in the simpler Pendulum system. With well-learned coordinates, models that enforce Lagrangian dynamics result in better long term prediction, e.g., as compared to MLPdyn+caVAE, since Lagrangian dynamics with zero control preserves energy (see Supplementary Materials).

**Learned potential energy enables energy-based control.** Figure 4 shows the learned potential energy of the three systems and reconstruction images at selected coordinates with Lagrangian+caVAE.

The learned potential energy is consistent with the true potential energy of those systems, e.g., the pendulum at the upward position has the highest potential energy while the pendulum at the downward position has the lowest potential energy. Figure 4 also visualizes the learned coordinates. Learning interpretable coordinates and potential energy enables energy-based controllers. Based on the learned encoding and dynamics, we are able to control Pendulum and fully-actuated Acrobot to the inverted position, and fully-actuated CartPole to a position where the pole points upward. The sequences of images of controlled trajectories as shown in Figure 3 are generated based on learned dynamics and encoding with Lagrangian+caVAE as follows. We first encode an image of the goal position $\mathbf{x}^\star$ to the goal generalized coordinates $\mathbf{q}^\star$. At each time step, the OpenAI Gym simulator of a system can take a control input, integrate one time step forward, and output an image of the system at the next time step. The control input to the simulator is $\mathbf{u}(\mathbf{q}, \dot{\mathbf{q}}) = \beta(\mathbf{q}) + \mathbf{v}(\dot{\mathbf{q}})$ which is designed as in Section 2.2 with the learned potential energy, input matrix, coordinates encoded from the output images, and $\mathbf{q}^\star$.

| Average pixel MSE | Pendulum (train | test) | | CartPole (train | test) | | Acrobot (train | test) | |
|---|---|---|---|---|---|---|
| **Lagrangian + caVAE** | 1.82 | 1.83 | **2.78** | **2.81** | **3.06** | **3.14** |
| Lagrangian + VAE | **1.52** | **1.56** | 9.41 | 10.30 | 10.48 | 10.78 |
| MLPdyn + caVAE | 1.92 | 1.92 | 14.18 | 14.97 | 12.12 | 12.14 |
| HGN | 0.55* | 0.71* | 2.81* | 2.91* | 2.61* | 2.67* |

Table 1: Average pixel MSE of different models. All values are multiplied by $1e+3$. The starred values are calculated from trajectories without control.

**Baselines** We set up two baseline models: HGN [6] and PixelHNN [3]. Neither model considers control input so we only use the trajectories with zero control in our dataset to train the models. Because of this, it is not fair to compare the pixel MSE of HGN in Table 1 with those of other models. We implemented HGN based on the architecture described in the paper and used the official code for PixelHNN. From Figure 3, we can see that HGN makes good predictions for the pendulum up to the training sequence length ($\tau = 4$), but makes blurry long term predictions. HGN fails to generalize to the test dataset for the CartPole task. This is probably because HGN is not data efficient. In the original HGN experiment [6], $30 \times 50K = 1.5M$ training images are used in the pendulum task, while here we use $20 \times 256 = 5120$ training images (with zero control). Moreover, the dimension of latent coordinate $\mathbf{q}$ is $4 \times 4 \times 16 = 256$. With such a fixed high dimension (for various tasks), HGN does not need to assume degrees of freedom. However, it might not be easy to interpret the learned $\mathbf{q}$, whereas the learned coordinates in our model are interpretable. PixelHNN does not use an integrator and requires a special term in the loss function. PixelHNN does not account for the rotational nature of coordinate $q$, so the reconstruction images around the unstable equilibrium point are blurry and the learned coordinates are not easy to interpret (see Supplementary Material). In the original implementation of PixelHNN [3], the angle of the pendulum is constrained to be from $-\pi/6$ to $\pi/6$, where a linear approximation of the nonlinear dynamics is learned, which makes the learned coordinates easy to interpret. This constraint does not hold for more challenging control tasks.

**Ablation study** To understand which component in our model contributes to learning interpretable generalized coordinates the most, we also report results of four ablations, which are obtained by (a) replacing the coordinate-aware encoder with a black-box MLP, (b) replacing the coordinate-aware decoder with a black-box MLP, (c) replacing the coordinate-aware VAE with a coordinate-aware AE, and (d) a Physics-as-inverse-graphics (PAIG) model [14]. We observe that the coordinate-aware decoder makes the primary contribution to learning interpretable coordinates, and the coordinate-aware encoder makes a secondary contribution. The coordinate-aware AE succeeds in Pendulum and Acrobot tasks but fails in the CartPole task. PAIG uses AE with a neural network velocity estimator. We find that PAIG's velocity estimator overfits the training data, which results in inaccurate long term prediction. Please see Supplementary Materials for prediction sequences of the ablation study.

## 5  Conclusion

We propose an unsupervised model that learns planar rigid-body dynamics from images in an explainable and transparent way by incorporating the physics prior of Lagrangian dynamics and a coordinate-aware VAE, both of which we show are important for accurate prediction in the image space. The interpretability of the model allows for synthesis of model-based controllers.

## Broader Impact

We focus on the impact of using our model to provide explanations for physical system modelling. Our model could be used to provide explanations regarding the underlying symmetries, i.e., conservation laws, of physical systems. Further, the incorporation of the physics prior of Lagrangian dynamics improves robustness and generalizability for both prediction and control applications.

We see opportunities for research applying our model to improve transparency and explanability in reinforcement learning, which is typically solved with low-dimensional observation data instead of image data. Our work also enables future research on vision-based controllers. The limitations of our work will also motivate research on unsupervised segmentation of images of physical systems.

## Acknowledgments and Disclosure of Funding

This research has been supported in part by ONR grant #N00014-18-1-2873 and by the School of Engineering and Applied Science at Princeton University through the generosity of William Addy '82. Yaofeng Desmond Zhong would like to thank Christine Allen-Blanchette, Shinkyu Park, Sushant Veer and Anirudha Majumdar for helpful discussions. We also thank the anonymous reviewers for providing detailed and helpful feedback.

## Footnotes

[1]Please see Supplementary Materials for more details.

[2]Here we expect to encode the angle of the pole from a pole image regardless of where it appears in the image. As the translation invariance of CNN is shown by Kauderer-Abrams [31] to be primarily dependent on data augmentation, the encoding of generalized coordinates might not generalize well to unseen trajectories. Also, in general we need both translation invariance and rotation invariance, a property that CNN do not have.

[3]We use the reparametrization trick proposed by Davidson et al. [30] to sample from a von Mises distribution.

[4]For a transformation that depends on one or more generalized coordinate, those generalized coordinates must be encoded before the transformation can be applied. In the CartPole example, we need to encode $r$ before applying the transformation to put the attention window centered at the cart to encode $\phi$. We use the mean of the distribution, i.e., $q_j^m$ or $(\cos q_j^m, \sin q_j^m)$, for those transformations that depend on $q_j$.

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
