[Supplementary Material]

# Unsupervised Learning of Lagrangian Dynamics from Images for Prediction and Control (Supplementary Material)

**Yaofeng Desmond Zhong**
Princeton University
y.zhong@princeton.edu

**Naomi Ehrich Leonard**
Princeton University
naomi@princeton.edu

## S1   Summary of model assumptions

Here we summarize all our model assumptions and highlight what are learned from data.

Model assumptions:

- **The choice of coordinates**: we choose which set of coordinate we want to learn and design coordinate-aware VAE accordingly. This is important from an interpretability perspective. Take the Acrobot as an example, the set of generalized coordinates that describes the time evolution of the system is not unique (see Figure 5 in [1] for another choice of generalized coordinates.) Because of this non-uniqueness, without specifying which set of coordinates we want to learn will let the model lose interpretability.

- **Geometry**: this includes the using von Mises distribution for rotational coordinates and spatial transformer networks for obtaining attention windows.

- **Time evolution of the rigid-body system can be modelled by Lagrangian dynamics**: since Lagrangian dynamics can modelled a broad class of rigid-body systems, this assumption is reasonable and help us gain interpretability.

What are learned from data:

- **Values of the chosen coordinates**: although we choose which set of coordinates to learn, the values of those coordinates cooresponding to each image are learned.

- **Shape, length and mass of objects**: the coordinate-aware decoder learns the shape of the objects first and place them in the image according to the learned coordinates. The length of objects are modelled as learnable parameters in the model when those length need to be used (e.g. Acrobot). The mass of the objects are learned as $\mathbf{M}(\mathbf{s}_1, \mathbf{s}_2, \mathbf{s}_3)$.

- **Potential energy**: this is learned as $V(\mathbf{s}_1, \mathbf{s}_2, \mathbf{s}_3)$.

- **How control influences the dynamics**: this is learned as $\mathbf{g}(\mathbf{s}_1, \mathbf{s}_2, \mathbf{s}_3)$.

## S2   Conservation of energy in Lagrangian dynamics

In the following, we review the well known result from Lagrangian mechanics, which shows that with no control applied, the latent Lagrangian dynamics conserve energy, see, e.g., Goldstein et al. [2], Hand and Finch [3].

**Theorem 1** (Conservation of Energy in Lagrangian Dynamics). *Consider a system with Lagrangian dynamics given by Equation (3). If no control is applied to the system, i.e, $\mathbf{u} = \mathbf{0}$, then the total system energy $E(\mathbf{q}, \dot{\mathbf{q}}) = T(\mathbf{q}, \dot{\mathbf{q}}) + V(\mathbf{q})$ is conserved.*

*Proof.* We compute the derivative of total energy with respect to time and use the fact that, for any real physical system, the mass matrix is symmetric positive definite. We compute

$$
\frac{\mathrm{d}E(\mathbf{q}, \dot{\mathbf{q}})}{\mathrm{d}t} = \frac{\partial E}{\partial \mathbf{q}}\dot{\mathbf{q}} + \frac{\partial E}{\partial \dot{\mathbf{q}}}\ddot{\mathbf{q}}
$$
$$
= \frac{1}{2}\dot{\mathbf{q}}^T \frac{\mathrm{d}\mathbf{M}(\mathbf{q})}{\mathrm{d}t}\dot{\mathbf{q}} + \dot{\mathbf{q}}^T \frac{\mathrm{d}V(\mathbf{q})}{\mathrm{d}\mathbf{q}} + \dot{\mathbf{q}}^T \mathbf{M}(\mathbf{q})\ddot{\mathbf{q}}
$$
$$
= \dot{\mathbf{q}}^T \mathbf{g}(\mathbf{q})\mathbf{u},
$$

where we have substituted in Equation (3). Thus, if $\mathbf{u} = \mathbf{0}$, the total energy $E(\mathbf{q}, \dot{\mathbf{q}})$ is conserved. □

With Lagrangian dynamics as our latent dynamics model, we automatically incorporate a prior of energy conservation into physical system modelling. This explains why our latent Lagrangian dynamics result in better prediction, as shown in Figure 3.

This property of energy conservation also benefits the design of energy-based controllers $\mathbf{u}(\mathbf{q}, \dot{\mathbf{q}}) = \beta(\mathbf{q}) + \mathbf{v}(\dot{\mathbf{q}})$. With only potential energy shaping $\beta(\mathbf{q})$, we shape the potential energy so that the system behaves as if it is governed by a desired Lagrangian $L_d$. Thus, the total energy is still conserved, and the system would oscillate around the global minimum of the desired potential energy $V_d$, which is $\mathbf{q}^\star$. To impose convergence to $\mathbf{q}^\star$, we add damping injection $\mathbf{v}(\dot{\mathbf{q}})$. In this way, we systematically design an interpretable controller.

## S3    Experimental setup

### S3.1    Data generation

All the data are generated by OpenAI Gym simulator. For all tasks, we combine 256 initial conditions generated by OpenAI Gym with 5 different constant control values, i.e., $u = -2.0, -1.0, 0.0, 1.0, 2.0$. For those experiments with multi-dimensional control inputs, we apply these 5 constant values to each dimension while setting the value of the rest of the dimensions to be 0. The purpose is to learn a good $\mathbf{g}(\mathbf{q})$. The simulator integrates 20 time steps forward with the fourth-order Runge-Kutta method (RK4) to generate trajectories and all the trajectories are rendered into sequences of images.

### S3.2    Model training

**Prediction time step $T_{\mathbf{pred}}$.**    A large prediction time step $T_{\text{pred}}$ penalizes inaccurate long term prediction but requires more time to train. In practice, we found that $T_{\text{pred}} = 2, 3, 4, 5$ are able to get reasonably good prediction. In the paper, we present results of models trained with $T_{\text{pred}} = 4$.

**ODE Solver.**    As for the ODE solver, it is tempting to use RK4 since this is how the training data are generated. However, in practice, using RK4 would make training extremely slow and sometimes the loss would blow up. It is because the operations of RK4 result in a complicated forward pass, especially when we also use a relatively large $T_{\text{pred}}$. Moreover, since we have no access to the state data in the latent space, we penalize the reconstruction error in the image space. The accuracy gained by higher-order ODE solvers in the latent space might not be noticable in the reconstruction error in the image space. Thus, during training, we use an first-order Euler solver. As the Euler solver is inaccurate especially for long term prediction, after training, we use RK4 instead of Euler to get better long term prediction results with learned models.

**Data reorganization.**    As our data are generated with 20 times steps in each trajectory, we would like to rearrange the data so that each trajectory contains $T_{\text{pred}} + 1$ time steps, as stated in Section 3. In order to utilize the data as much as possible, we rearrange the data $((\tilde{\mathbf{x}}^1, \mathbf{u}^c), (\tilde{\mathbf{x}}^2, \mathbf{u}^c), ..., (\tilde{\mathbf{x}}^{20}, \mathbf{u}^c))$ into $((\tilde{\mathbf{x}}^i, \mathbf{u}^c), (\tilde{\mathbf{x}}^{i+1}, \mathbf{u}^c), ..., (\tilde{\mathbf{x}}^{i+T_{\text{pred}}}, \mathbf{u}^c))$, where $i = 1, 2, ..., 20 - T_{\text{pred}}$. Now the length of each reorganized trajectory is $T_{\text{pred}} + 1$.

**Batch generation.**    There are two ways of constructing batches of such sequences to feed the neural network model. The standard way is to randomly sample a certain number of sequences from the reorganized data. An alternative way is to only sample sequences from a specified constant control

so that within each batch, the control values that are applied to the trajectories are the same. Of course, this control value would change from batch to batch during training. The reason behind this homogeneous control batch generation is the following. Except $\mathbf{g}(\mathbf{s}_1, \mathbf{s}_2, \mathbf{s}_3)$, all the vector-valued functions we parametrized by neural networks can be learned by trajectory data with zero control. From this perspective, all the trajectory data with nonzero control contributes mostly to learning $\mathbf{g}(\mathbf{s}_1, \mathbf{s}_2, \mathbf{s}_3)$. If this is the case, we can feed trajectory data with zero control more frequently and all the other data less frequently. All the results shown in the main paper and in the next section are trained with homogeneous control batch generation. We present the results of the standard batch generation in Section S5.

**Annealing on weight $\lambda$.** In Equation (16), the vM regularization with weight $\lambda$ penalizes large norms of vectors $(\alpha_j, \beta_j)$. In our experiments, we found that sometimes an annealing scheme such as $\lambda = \min(i_e/8000, 0.375)$ results in better reconstruction images than a constant weight, where $i_e$ is the current epoch number during training.

**Optimizer.** For all the experiments, we use the Adam optimizer to train our model.

## S4   Ablation study details

We report on the following four ablations:

(a) **tradEncoder + caDecoder**: replacing the coordinate-aware encoder with a traditional black-box MLP

(b) **caEncoder + tradDecoder**: replacing the coordinate-aware decoder with a traditional black-box MLP

(c) **caAE**: replacing the coordinate-aware VAE with a coordinate-aware AE

(d) **PAIG**: a Physics-as-inverse-graphics model

Figure 5 shows the prediction sequences of ablations of Pendulum and CartPole. Our proposed model is labelled as caVAE. Since long term prediction of the chaotic Acrobot is not possible, Figure 6 shows the reconstruction image sequences of ablations of Acrobot. From the results, we find that PAIG and caAE completely fails in CartPole and Acrobot, although they work well in the simple Pendulum experiment. By replacing the coordinate-aware decoder, caEncoder+tradDecoder fails to reconstruct rigid bodies in CartPole and Acrobot. By replacing the coordinate-aware encoder, tradEncoder+caDecoder reconstructs correct images with well-learned coordinates in Pendulum and Acrobot, but in CartPole, the coordinates are not well-learned, resulting in bad prediction. Thus, we conclude that the coordinate-aware decoder makes the primary contribution to learning interpretable generalized coordinates and getting good reconstruction images, while the coordinate-aware encoder makes a secondary contribution.

## S5   Results with standard batch generation

From Figure 3 and Figure 7, we can see that our model behaves consistently well in prediction. In our experiments, we observe that our model trained with standard batch generation fail to learn a good model in the Acrobot example for the purpose of control. How the batch generation and annealing scheme affect the quality of learned model is not easy to investigate. We leave this to future work.

## S6   Blurry reconstruction of PixelHNN

The original PixelHNN are trained with sequences of pendulum image data limited to a small angle range $[-\pi/6, \pi/6]$. Here we train the PixelHNN with randomly sampled angles as initial conditions. Figure 8 and Figure 9 shows the prediction results and the latent trajectory of image sequences. As we can see, the latent space does not have an interpretable shape as in the original implementation (see cell 6 at this link). The prediction sequences are blurry especially near the unstable equilibrium point.

Figure 5: Prediction sequences of ablations of Pendulum and CartPole with a previously unseen initial condition and zero control. For the Pendulum experiment, the coordinate-aware encoder is a traditional MLP encoder. All the ablations get good predictions. For the CartPole experiment, all the ablations fail to get good predictions. The *PAIG* is able to reconstruct the cart initially but it fails to reconstruct the pole and make prediction. The *caAE* fails to reconstruct anything. The *caEncoder+tradDecoder* fails to reconstruct meaningful rigid bodies. The *tradEncoder+caDecoder* seems to extract meaningful rigid bodies but it fails to put the rigid bodies in the right place in the image, indicating the coordinates are not well learned.

Figure 6: Reconstruction image sequences of ablations of Acrobot with a previously unseen initial condition and zero control. The *PAIG* and *caAE* fail to reconstruct anything. The *caEncoder+tradDecoder* fails to reconstruct the green link at all. The *tradEncoder+caDecoder* makes good reconstruction.

Figure 7: Prediction sequences of Pendulum and CartPole with a previously unseen initial condition and zero control, trained with standard batch generation, as explained in Section S3.2.

Figure 8: **Left**: True sequence and PixelHNN prediction of an initial position of the pendulum far away from the unstable equilibrium point. **Right**, the latent coordinate $z$ and its time derivative $\dot{z}$ of the PixelHNN prediction sequence.

Figure 9: **Left**: True sequence and PixelHNN prediction of an initial position of the pendulum close to the unstable equilibrium point. **Right**, the latent coordinate $z$ and its time derivative $\dot{z}$ of the PixelHNN prediction sequence.