[Reviews · NeurIPS 2020]

Review 1

Summary and Contributions: The authors propose a VAE coordinate aware to learn Lagrangian dynamics. The architecture assumes that the number of (rigid) objects in an image are given and segmented. The encoder learns generalized coordinates from image/control pairs. The latent coordinates are modeled using Gaussian (for translational) and vonMises distributions (for rotational). The approach was tested with 3 types of dynamics: a pendulum, a cart-pole and a acrobot.

Strengths: The problem set up is well done. Learning physical dynamics from image/control pairs is novel and interesting. It uses first principles and hence the resulting architecture is explainable. The experiments have ablation studies showing the contribution of the modules of the model.

Weaknesses: It does presume that the number of objects is given and that they have been segmented; the examples are of simple dynamics. Experiments are qualitative rather than quantitative. Figure 3, second row shows that even for the simplest case (pendulum) the synthesis are not quite what they should be (frames 3, 4, 5, 8, 9, and 10 are noticeable different from the true sequence frames)

Correctness: As far as I can tell, yes they are correct. However I'd have liked to see quantitative results as well.

Clarity: Yes.

Relation to Prior Work: Yes.

Reproducibility: Yes

Additional Feedback: I'd like to thank the authors for the rebuttal and the provided experiment. I've upgraded my ranking.


Review 2

Summary and Contributions: The paper presents a way of learning the lagrangian dynamics for rigid body systems from images. Similarly to recent work (Greydanus'19, Toth'20), this is done with a NeuralODE approach wherein the state of the system (in generalized coordinates) is inferred from an image with a neural network, and this representation is learned end-to-end with a prediction loss. It is shown that the resulting system learns interpretable potential energies and can be used for prediction and control in continuous dynamics settings. --- Decision --- The paper presents an interesting and potentially promising method for learning Lagrangian dynamics from images. However, prior work exists that already solves a very similar problem for Hamiltonian dynamics, some of which is not referenced in the paper (Toth'20). These baselines are not evaluated. Moreover, the paper does not specify neither the contributions nor the motivation behind the work. Unfortunately, I am not able to accept this paper since the importance of solving this problem for Lagrangian rather than Hamiltonian networks is not explained and the paper does not compare to prior work. Toth'20, HAMILTONIAN GENERATIVE NETWORKS ---- Update ---- The authors' response and the internal discussion have clarified many of my questions about the paper, and I raise my rating to borderline. Further, I believe the new changes will improve the paper significantly after they are incorporated. However, I believe that since significant changes in writing and more experiments are needed to correctly position the paper with respect to the prior work, the paper should undergo another review round after the updates. I provide my detailed reasoning and feedback below. ---- Positioning I believe that the paper could have the following key contributions: image-based lagrangian dynamics for prediction, interpretability of the latent rigid body coordinates, and image-based lagrangian dynamics for control. Unfortunately, current paper does not specify which contribution they are claiming, and does not convincingly defend any of these contributions. The authors may wish to focus on one or multiple of these contributions and defend it by proper theoretical and empirical comparison. Given the promising results in the rebuttal, I suspect that the next iteration of the paper can successfully argue for the first contribution by discussing and comparing to the baselines discussed in the rebuttal. Alternatively, the authors may wish to focus on the second contribution, in which case it is crucial to extensively discuss how prior work relates to the method, provide baseline comparisons beyond ablations, and provide quantitative metrics that show that the method outperforms the baselines and ablations. Should the authors wish to focus on third contribution, I would expect a discussion of why energy shaping control is beneficial to the mainstream visual control methods. ---- Additional comments The abstract and perhaps the title needs to clarify that the Lagrangian of a rigid body system is learned. This is not obvious since recent work has applied Hamiltonian/Lagrangian to general latent spaces as opposed to rigid body systems. The introduction talks about learning physical dynamics with neural networks. Rather confusingly, it entirely omits references to the mainstream methods that do so, such as Ebert'18 or Hafner'19. It is crucial to clarify in the introduction that this paper focuses on interpretable dynamics models for rigid body systems instead of general learned physics. Re motivation: I am glad the authors have added some motivation referencing the results in the rebuttal. I am hopeful that once incorporated in the main paper, this motivation will be clear to the future reader.

Strengths: The paper proposes an interesting approach to learning Lagrangian dynamics for rigid body systems from images.

Weaknesses: The paper does not specify its contributions and there is no motivation provided. The introduction mentions that prior Hamiltonian neural networks have not been applied to image data, but this is not true (see [6] or Toth'20). These baselines are missed. The experimental section does not explain why the experiments were performed, and the conclusion does not discuss the impact of the paper.

Correctness: A nitpick: the paper states (l163) that "a black-box neural network would not be able to learn interpretable generalized coordinates for a system in motion described by Lagrangian dynamics." This is not true in theory since neural networks are universal approximators, and hard to believe in practice since this task is only a very simple keypoint detection task, while current computer vision systems are able to handle much more complex data, e.g. see Cao'19. Cao'19, OpenPose: Realtime Multi-Person 2D Pose Estimation using Part Affinity Fields

Clarity: The method is clearly explained, but the exposition is a bit haphazard. A large part of the method is somewhat unexpectedly presented in the background section. The methods section does not contain a formal overview of the model (except for Figure 1, which is very hard to read), which makes it hard to parse. The introduction and experimental sections are missing motivation and explanation of contributions.

Relation to Prior Work: The paper misrepresents existing work on Hamiltonian neural networks Greydanus'19, Toth'20), claiming that prior work does not handle image observations. Some of this relevant work is not cited (Toth'20). The paper does not explain how the proposed approach is different and these baselines are not evaluated.

Reproducibility: Yes

Additional Feedback: It seems quite unexpected that the system generalizes to novel control inputs while only being shown constant control at training time. How is this achieved? Why not train with variable control inputs? The paper uses a time-varying mass matrix and a time-varying input matrix, which is likely to cause the mass matrix become unidentifiable (since an arbitrary motion can be produced by changing the mass matrix in time), and the controls to possibly have different interpretations at different times. What is the reason to do this?


Review 3

Summary and Contributions: This paper builds on a recent line of work that enforces Lagrangian and Hamiltonian dynamics on neural network-based models of dynamical systems. The authors consider the case where coordinates are embedded in high-dimensional data such as images, show how to learn a Lagrangian for such systems, and then use them to solve control problems. Their empirical results represent a significant contribution. The paper is candidly written, has good ablation studies, and is easy to follow. ### Note: I have updated this review to take author feedback and reviewer discussion into account. See "Additional feedback" for details.

Strengths: - The writing, equations, and figures are clear and coherent. - The empirical tasks are challenging and relevant to the NeurIPS community. - The authors tackle a significant problem which is underexplored by the community. - The method is sound and general; it incorporates strong physics priors but is still flexible enough to do well in three disparate domains (pendulum, cartpole, acrobot) - The authors do a good job of placing this paper in the broader context of recent work on learning Lagrangians and Hamiltonians from data (with one exception, see next section)

Weaknesses: - The authors do not acknowledge work by Toth et al 2019 (arxiv.org/abs/1909.13789) which learns dynamics from pixels (using Hamiltonians rather than Lagrangians). Seems extremely relevant. To be clear: I am not one of the authors on that paper; I am making this note because I think it’s a potential blind spot of this paper - Equation 2 makes a simplifying assumption such that this approach only works only for rigid body systems. - It's unclear how well the coordinate-aware encoder will generalize to real-world systems. Although showing such applications is beyond the scope of this paper, I would like to see some discussion from the authors on how they believe this might take place. To the authors: in what ways do you see this new technique - the ability to learn Lagrangians from pixels - being helpful in applied settings? - There are several important assumptions embedded in this model. The first is that the coordinates are rotational (there are periodic activation functions on them). The second is the coordinate transformation function learned by the STN, which one can restrict or relax according to the problem in question (lines 194-200). The third is that the Lagrangian is that of a rigid body system (line 65). These assumptions appear to be sufficiently general. However, I’d like to hear the authors discuss these assumptions, and add any that I’ve missed here (To the authors and other reviewers: are these assumptions sufficiently general?). Explicitly stating all physical priors/assumptions embedded into this model in one place would improve this work.

Correctness: The equations, figures, text, and methods are sound. Good work.

Clarity: The clarity of the writing, equations, and figures is all well above average. Congratulations to the authors for work well done. This paper was a pleasure to read.

Relation to Prior Work: - Lines 23-27 claim “Another class of approaches learn physical models from images, by either learning the map from images to coordinates with supervision on true coordinate data [9] or learning the coordinates in an unsupervised way but only with translational coordinates [10, 11]. The unsupervised learning of rotational coordinates such as angles of objects are under-explored in the literature.” This is not entirely true. Both (Greydanus et al 2019) and (Toth et al 2019) were able to learn Hamiltonian dynamics from pixel images. This paper does not discuss those closely related contributions, and this is one of its main shortcomings. - In comparison to (Greydanus et al 2019) and (Toth et al 2019), this paper *does* make significant progress. But in order to make that progress clear, it would help for the authors to provide some discussion so as to make clear the ways in which this approach represents an improvement. - Discussion of other related work is clear, correct, and sensible.

Reproducibility: Yes

Additional Feedback: ####### Updates in light of author feedback and reviewer discussion ####### ### My initial response to the author feedback and other reviews - R1 asks for qualitative results and the authors made this change (line 16 of rebuttal). Good ask by R1, and good response from the authors. - I agree with R1 and R2 about misrepresentation of previous work. In both (Greydanus et al 2019) and (Toth et al 2020) the authors considered pixel tasks. Especially (Toth et al 2020) since there is currently no mention of it. This is essential to have in the camera-ready version. - I disagree with R2’s nitpick about line 163. The universal function approximator theorem is an overused argument, and in practice the architectural inductive biases of a neural network matter enormously. As a silly example, a *very* wide one-layer MLP should *theoretically* be able to match the performance of ResNet-152 on ImageNet, but by playing with infinite limits we obscure the fact that ResNet-152 is more practically effective because of its dramatically better inductive biases. - I like R4’s point about the importance of combining the learned Lagrangian with energy-based control. This, in fact, seems to be one of the most interesting and significant contributions of the paper. - Authors, line 47: Thanks for agreeing to add a paragraph about all assumptions to the appendix. This will improve the paper. - Based on the comments of the other reviewers and the authors’ response, I continue to be pleased with this paper overall and recommend acceptance. However, I agree with the other reviewers who posit that the paper does not explain its significance as well as it could. I have updated my score to a [7]. - I continue to recommend acceptance, as the contribution is significant even if not specifically stated. If this paper is not accepted to NeurIPS 2020, I hope the authors will make improvements to the text so that this interesting work will have the impact it deserves.


Review 4

Summary and Contributions: This work incorporates Lagrangian dynamics as a physical prior into neural networks for motion prediction and control. Variational autoencoders are used to infer rotational and translational coordinates from images.

Strengths: 1. The proposed model is technically sounding. Their framework mainly consists of VAE to extract coordinates, and a learned neural ODE to constrain the states to follow Lagrangian dynamics. This model can operate on raw images, compared with other works that require low-dimensional states. 2. Their model can encode important system properties such as potential energy, which can further allow energy based control for Pendulum and Acrobot. This point is very interesting and useful. 3. Their model enables long-term prediction up to 48 time steps.

Weaknesses: 1. In the experiment, they only compares with some baselines they created, but there is a lack of comparison with either existing Lagrangian dynamics network or VAE based prediction network. For example, Kalman VAE combines kalman filtering with VAE for prediction and control, which performs well in pendulum control and trajectory prediction from images. Their code is open-sourced. I would suggest to compare your method with it for a more convincing evaluation. 2. Some important experiment details are missing, such as the number of training/validation/testing samples, batch-size, learning rate, hidden states inside your framework. Also, the authors didn't report quantitative results. It's hard to convince me, with only two testing trajectories for prediction and one testing trajectory for control provided.

Correctness: Their claims and method are correct.

Clarity: This paper is well-written and easy-to-follow.

Relation to Prior Work: The authors should discuss relevant work on neural motion prediction and control from images, and claim the advantage of introducing Lagrangian dynamics into neural network in this task, as their main contribution is to use Lagrangian dynamics for this task.

Reproducibility: Yes

Additional Feedback: I have read the authors' response, and appreciate the authors' efforts on improving this paper. The topic of this paper is very interesting to me, and I am convinced that the contribution of this paper is significant. Especially, I like the energy-based control part, which is underexplored. But I still have concerns about its current form, including the comparison with other state-of-the-art, and demonstration of their main contributions. Thus, I would like to keep score unchanged.

[Author Response · NeurIPS 2020]

We thank reviewers for their thoughtful feedback! We are encouraged that they find our problem setup well-done (R1)
and significant (R3). We are glad they think our method is novel (R1), sound (R3, R4) and potentially promising (R2).
We are pleased that they find our method learns explainable (R1) and interpretable (R2) energy that allows for control
(R2, R4). We address a few specific concerns below and will incorporate all feedback in the camera-ready version.

@R2 - **Motivation:** Humans can learn to predict the trajectories of mechanical systems, e.g., basketballs or drones,
from high-dimensional visual inputs, and learn to control the system, e.g., catch a ball or hover a drone, after a small
number of interactions with those systems. We hypothesize that humans use domain-specific knowledge, e.g., physics
laws, to achieve efficient learning. Motivated by this hypothesis, we incorporate the Lagrangian prior to learn and
control dynamics from image data, hoping to gain interpretability and data efficiency (Please see L25-27 below.)

@R1, R4 - **Quantitative results:** We pointed out (L240) that pixel MSE is not a good metric for *long term* prediction.
Please see [10] (Sec. 1) for more discussion. Instead, in the table below, we report the *short term* ($T_{\text{pred}} = 4$) average
pixel MSE. Our proposed model has significantly better performance on the CartPole and Acrobot tasks than model
variants and has similar performance to model variants on the simpler pendulum task. To R4, we have reported on
training data generation including the sizes (SM Sec. S2.1). We generated testing data of equal size. We will include
details on model architecture in camera-ready. We will also open source our code for all the experiments.

| Average pixel MSE | Pendulum (train | test) | | CartPole (train | test) | | Acrobot (train | test) | |
|---|---|---|---|---|---|---|
| **Lagrangian + caVAE** | 1.82 | 1.83 | **2.78** | **2.81** | **3.06** | **3.14** |
| Lagrangian + VAE | **1.52** | **1.56** | 9.41 | 10.30 | 10.48 | 10.78 |
| MLPdyn + caVAE | 1.92 | 1.92 | 14.18 | 14.97 | 12.12 | 12.14 |

@R2, R3 - **Compare with prior work HNN [6], HGN [Toth'20]:** The main contribution of HNN is learning Hamil-
tonian dynamics from low dimensional data. Although they proposed PixelHNN to learn from pendulum images, the
angle of the pendulum in the training data is constrained to be in $[-\pi/6, \pi/6]$. From our experiments, PixelHNN
does not work for pendulum with large angles and does not generalize to CartPole and Acrobot. [Toth'20] also has a
similar observation about PixelHNN. We thank R2 and R3 for pointing to HGN [Toth'20], but we'd like to point out
three differences. 1) Control. Both HGN and HNN learn energy-conserved dynamics and do not learn control. It is
not clear how to incorporate control into HGN. 2) Interpretability. In HGN's pendulum task, the dimension of $q$ is
$4 \times 4 \times 16 = 256$. With such a high dimension (for various tasks), HGN does not assume the degree of freedom is
given, but it might not be easy to interpret the learned $q$, while ours are interpretable. 3) Data efficiency. HGN uses
$30 \times 50K = 1.5M$ training images for the pendulum task, while we use $20 \times 256 = 5120$ training images (with zero
control). We may have a huge advantage on data efficiency. Thus, we disagree with R2 that our work is *"solving ...*
*problem for Lagrangian rather than Hamiltonian ..."*. However, we agree with R2 and R3 that we should acknowledge
[Toth'20] and we will update inaccurate statements about prior works. We are working on implementing HGN and we
will report the comparison with HGN in terms of pixel MSE, interpretability and data efficiency in the camera-ready.

@R4 - **Compare with Kalman VAE (KVAE):** Thanks for pointing to KVAE. We are not convinced about *"(KVAE)...*
*performs well in pendulum control"*(R4). Although KVAE models control inputs, there is neither controller design nor
control results reported in the paper. No pendulum data or pendulum prediction sequences are provided in their Github
repo and project website. However, we will investigate KVAE trained with our own pendulum data in camera-ready.

@R4 - **Discuss neural motion prediction and control from images:** We will add the discussion in camera-ready.

@R1 - **Visual difference in Fig. 3:** Our dynamical model enforces energy conservation (Thm 1 in SM), so the learned
energy will not drift away from the real constant energy but will oscillate around it. This oscillation explains the visual
difference. This oscillation also shows up in prior works. Please see the oscillations in [6] (Fig. 2) and [7] (Fig. 6).

@R2 - **Clarity and exposition:** We presented self-contained preliminary concepts and methods that have been used
in the prior works in Sec. 2 to prevent a hard-to-follow Sec. 3, which already takes four pages. R3 and R4 say our
paper is *"a pleasure to read"* and *"easy to follow"*, respectively. We provided a short overview of Sec. 3 in L112-120.

@R2 - **Training with constant control:** We didn't elaborate since it has been explained in a prior work [7] (Sec. 3.1.)

@R2 - **Time-varying mass matrix and input matrix:** In general, the *mass matrix* is not constant even if each rigid
body has a constant mass, e.g., an Acrobot. Please see the appendix of Sutton'96 for the dependence of mass matrix on
the angle of the Acrobot. In general, the input matrix is also non-constant. The dependency on coordinates is necessary.
[Sutton'96] Generalization in Reinforcement Learning: Successful Examples Using Sparse Coarse Coding.

@R3 - **Assumptions in one place:** We will summarize and discuss all assumptions in the appendix in camera-ready.

@R3 - **Pros in applied settings:** We imagine this new technique would benefit a robot equipped with camera sensors
to learn to predict and control other systems (robots), because of the interpretability and data efficiency.

[Meta-Review · NeurIPS 2020]

This paper makes it possible to learn Lagrangian dynamics from images and use them for energy-based control. This represents an important and significant advance for this fledgling new research subfield of physics-aware prediction, which might very well go on to prove important and significant in the coming years. I believe the reviewers are all in agreement on this point. However, by entering this new territory for physics-aware prediction, this paper has also exposed itself to interest from a broader community of readers and NeurIPS attendees who are familiar with the progress in image-based *intuitive physics* modeling and control methods over the last 5 years or so (R2 and R4 point to some such approaches). A lot of the difficulty in arriving at a reviewer consensus for this paper can be put down to the fact that its positioning is somewhat myopic and ignores this broader context, perhaps because the authors themselves might not be familiar with these approaches. However, I would urge the authors to position their work within this broader image-based prediction and control, to help introduce not only their work but this family of approaches to a broader audience. For maximizing impact, this might even mean evaluating such methods as baselines (e.g. Ebert '18) --- note that you may not necessarily have to beat those methods given your other advantages; but these comparisons are important to place this approach in its appropriate context. Even within the physics-aware prediction literature, some references and comparisons and a clear statement of assumptions are missing, as is a clear statement of the claims, though the rebuttal has begun to address these. In particular, comparison to Toth '20 (in zero-control settings) is important. These above changes, had they already been made, would have allowed me to recommend a strong accept without many reservations. At present, I am only able to tentatively recommend a poster presentation, while urging the authors to incorporate these suggestions into their camera-ready.